# Undoped and Eu^3+^ Doped Magnesium-Aluminium Layered Double Hydroxides: Peculiarities of Intercalation of Organic Anions and Investigation of Luminescence Properties

**DOI:** 10.3390/ma12050736

**Published:** 2019-03-04

**Authors:** Aurelija Smalenskaite, Lina Pavasaryte, Thomas C. K. Yang, Aivaras Kareiva

**Affiliations:** 1Department of Inorganic Chemistry, Vilnius University, Naugarduko 24, LT 03225 Vilnius, Lithuania; aivaras.kareiva@chgf.vu.lt; 2Department of Chemical Engineering and Biotechnology, National Taipei University of Technology, 1, Sec. 3, Chung-Hsiao E. Road, Taipei 106, Taiwan; lina.pavasaryte@gmail.com (L.P.); ckyang@ntut.edu.tw (T.C.K.Y.); 3Center for Precision Analysis and Materials Research, National Taipei University of Technology, 1, Sec. 3, Zhongxiao E. Rd., Taipei 10608, Taiwan

**Keywords:** LDHs, Eu doping effect, intercalation of organic species, size effect, luminescence

## Abstract

The Mg_3_/Al and Mg_3_/Al_0.99_Eu_0.01_ layered double hydroxides (LDHs) were fabricated using a sol-gel chemistry approach and intercalated with different anions through ion exchange procedure. The influence of the origin of organic anion (oxalate, laurate, malonate, succinate, tartrate, benzoate, 1,3,5-benzentricarboxylate (BTC), 4-methylbenzoate (MB), 4-dimethylaminobenzoate (DMB) and 4-biphenylacetonate (BPhAc)) on the evolution of the chemical composition of the inorganic-organic LDHs system has been investigated. The obtained results indicated that the type and arrangement of organic guests between layers of the LDHs influence Eu^3+^ luminescence in the synthesized different hybrid inorganic–organic matrixes. For the characterization of synthesis products X-ray diffraction (XRD) analysis, infrared (FTIR) spectroscopy, fluorescence spectroscopy (FLS), and scanning electron microscopy (SEM), were used.

## 1. Introduction

A general chemical formula of layered double hydroxides (LDHs) is [M^2+^_1−*x*_M^3+^*_x_*(OH)_2_]*^x^*^+^(A*^y^*^−^)*_x_*_/*y*_·*z*H_2_O, here M^2+^ and M^3+^ are divalent and trivalent cations forming layered structure, respectively, and A*^y^*^−^ is anion occupying interlayer space [1]. LDHs show hexagonal crystal structure that depends on different parameters of the intercalated species. Intercalation of different anions in LDH is a challenging topic because the anion-exchange could be performed mostly, when the introduced anion has higher affinity with the LDH layer than the host anion. Usually, the anions with small size and high charge density are used for such investigations. Nevertheless, the low-charge large organic anions could also be introduced to the LDH structure [2]. The possibility to substitute of monovalent anions in the Mg/Al LDH could be expressed by following order OH^−^ > F^−^ > Cl^−^ > Br^−^ > NO^3−^. More selective are anions with higher charge CO_3_^2−^ > SO_4_^2−^ [3]. The anion-exchange selectivity is usually related to the guest orientation. Two orientations are observed for the organic anion within the gallery either vertical perpendicular to the layers or horizontal. Whether a vertical or horizontal orientation exists, depends upon the charge on the layers and the degree of hydration of the sample. Moreover, the water molecules stabilize the LDH structure via formation of a hydrogen bond [4,5,6]. The organic anions can create negative charge in the LDH particles, which can be associated to the micellization or formation of self-assembly of exchanged or adsorbed organic anions on the LDH surface [7]. Furthermore, interaction between LDHs carbonate and carboxylate-containing substances is an important aspect of the high affinity of these types of anions to the LDH surface. The adsorption of monovalent anions on the positively charged surface can proceed differently, which can be classified by the indirect Hofmeister series of the ions. Deviation from the series of the ions was observed only for the HCO_3_^−^ ions due to pronounced very high affinity to the LDH materials. Multivalent anions exhibit also high affinity to the LDH surfaces neutralizing the charge of the surface or even making it reversal at higher concentrations. This feature is more pronounced for anions with higher negative charge and platelets of significant negative charge could be formed. These results allow one to design LDH-based ion-exchange systems for different applications [8,9,10]. Besides, LDH intercalated with amino, ethylenediaminetetraacetic, diethylenetriaminepentaacetic, citric and malic acids could be used as adsorbents to remove toxic cationic and anionic species from aqueous media [11]. Moreover, LDHs intercalated with succinic acid and lauric acid were used as lubricant additives [12].

In recent years, inorganic-organic hybrid luminescence materials have been widely investigated due to the novel properties to form stable compounds with lanthanides based on unique anion exchange ability in the interlayer space of LDH. New photoluminescence materials when LDHs were doped with rare-earth (RE) have been synthesized [13,14]. These multifunctional materials are useful in many fields such as medicine, photochemistry, catalysis, environmental applications [15,16]. However, these LDHs show limitation due to the low intensity of emission which is caused by direct coordination of water molecules and hydroxyl groups to the RE centre in the layer. For the hybrid RE-organic LDH materials, intercalation of guest organic anions in the interlayer galleries influences the luminescence properties dramatically. For example, the enhanced green luminescence for Tb^3+^ by terephthalate anions in Tb-doped LDH was observed [17]. It was determined the possible energy transfer from the excited state of the introduced anion to Tb^3+^ active centres. The organic anions or neutral molecules should be chemically stable and have good solubility, significant mobility and capability to form amorphous layers [18]. The organic groups which have different donor-acceptor capabilities, different size and different lability are introduced into the LDHs host position. In the previous reports, the luminescence properties of Eu^3+^-doped LDHs intercalated by certain organic compounds, such as naphtalene-1,5-sulfonate, naphtalene-2,6-dicarboxylate [19] citrate, glutamate, picolinate, ethylenediaminetetraacetate [20], and many other compounds [21,22,23,24] have been investigated and discussed.

The goal of the present study was to investigate the luminescence properties of the Eu^3+^ doped LDHs containing organic anions. The influence of the origin of organic anion (oxalate, laurate, malonate, succinate, tartrate, benzoate, 1,3,5-benzentricarboxylate (BTC), 4-methylbenzoate (MB), 4-dimethylaminobenzoate (DMB) and 4-biphenylacetonate (BPhAc)) on the evolution of the chemical composition of the inorganic–organic LDHs system has been investigated.

## 2. Experimental

### 2.1. Synthesis of LDHs

The Mg_3_/Al and Mg_3_/Al_0.99_Eu_0.01_ LDH specimens were prepared by sol-gel technique using metal nitrates Mg(NO_3_)_2_·6H_2_O, Al(NO_3_)_3_·9H_2_O and Eu(NO_3_)_3_·6H_2_O, dissolved in 50 mL of deionized water as starting materials. To the obtained mixture, the 0.2 M solution of citric acid was added. The resulted solution was additionally stirred for 1 h at 80 °C. Finally, 2 mL of ethylene glycol was added with continued stirring at 150 °C. During the evaporation of solvent, the transformations from sol to the gel occurred. The synthesized precursor gel was dried at 105 °C for 24 h and was used for the synthesis of LDHs. The Mg_3_/Al and Mg_3_/Al_0.99_Eu_0.01_ LDHs were fabricated by reconstruction of mixed-metal oxides (MMO) in deionized water at 80 °C for 6 h, The MMO were formed during annealing the gels at 650 °C for 4 h.

### 2.2. Intercalation of Mg_3_/Al and Mg_3_/Al_0.99_Eu_0.01_ LDHs with Organic Anions

Mg_3_/Al or Mg_3_/Al_0.99_Eu_0.01_ benzoate, oxalate, laurate, malonate, succinate, tartrate, 1,3,5-benzentricarboxylate (BTC), 4-methylbenzoate (MB), 4-dimethylaminobenzoate (DMB) and 4-biphenylacetonate (BPhAc) were synthesized using anion exchange technique. For this, 2 mmol of Mg_3_/Al or Mg_3_/Al_0.99_Eu_0.01_ was immersed in the solution of disodium/sodium organic compounds with 1.5 molar excess amounts in comparison with LDHs. Next, the solution was stirred at room temperature for 24 h. After filtration and washing with deionized water and acetone, the synthesis product was dried at 40 °C for 12 h.

### 2.3. Characterization

X-ray diffraction analysis (XRD, Rigaku Mini Flex, Rigaku, The Woodlands, TX, USA) of synthesized compounds were performed with MiniFlex II diffractometer (Rigaku) using a primary beam Cu Kα radiation (λ = 1.541838 Å). The 2θ angle of the diffractometer was gradated from 8 to 80° in steps of 0.02°, with the measuring time of 0.4 s per step. Fourier-transform infrared spectroscopy (FT-IR) spectra were recorded using Bruker-Alpha FT-IR spectrometer (Bruker, Ettlingen, Germany) in the range of 4000–400 cm^−1^. The luminescent properties were investigated using Edinburg Instruments FLS 980 spectrometer (Edinburgh Instruments, Kirkton Campus, UK). The surface morphological features were characterized using a scanning electron microscope (SEM, Hitachi, Tokyo, Japan) Hitachi SU-70. The particle and anion dimension sizes were calculated using the ImageJ and Avogadro programmes (Jolla, CA, USA). The amount of carbonate in the synthesized samples was calculated from the M^II^/M^III^ atomic ratios, assuming that carbonate is the only charge balancing interlayer anion. The water content in the formula was determined from the results of TG analyses. The chemical composition was defined to be [Mg_0.75_Al_0.25_(OH)_2_] (CO_3_)_0.125_·4H_2_O.

## 3. Results and Discussion

It is reported [3] that LDH containing not only nitrates or chlorides, but also CO_3_^2−^ could be used for intercalation of other inorganic anions. Free CO_3_^2−^ and the NO_3_^−^ anions show similar symmetry, however, behave differently as interlayer anions in LDHs structure. The CO_3_^2−^ is orientated parallel to the hydroxide layers. It can easily interact with hydroxyl groups of hydroxide layers by forming hydrogen bonds [25]. The NO_3_^−^ has molecular plane tilted orientation, which makes disorder of the 3R rhombohedral symmetry [26] within a hexagonal unit cell of LDH crystal structure. Previously, the LDHs were obtained using the anion-exchanged method showing that values of basal spacing *c* increased significantly in comparison with starting carbonate containing LDH [11]. The parameter *c* depends on the size, charge and orientation of the intercalated species.

In this work, the intercalated organic anions, such as short-long carbon chains (oxalate, laurate, malonate, succinate, tartrate) and benzoic (benzoate, 1,3,5-benzentricarboxylate, 4-methylbenzoate, 4-dimethylaminobenzoate and 4-biphenylacetonate) carboxyl acid groups could be arranged by anions size in the interlayer and by the charge to compensate the hydroxide layer. In the XRD patterns of the LDH phases obtained by the anion exchanged reactions the diffraction peaks were shifted to the lower values of 2θ angle proving that values of the basal spacing *c* increased. The positions of diffraction peaks (003) of LDHs intercalated with short-long chains (Mg_3_/Al-succinate, Mg_3_/Al-malonate, Mg_3_/Al-tartrate, Mg_3_/Al-laurate and Mg_3_/Al-oxalate (see Figure 1)) are shifted to smaller 2θ angle values. The similar shift was observed and for the LDHs modified with benzoic carboxylates (Figure 2). The determined values of the lattice parameters *c* (see Table 1) were monotonically increased from *c* = 23.613 Å for the Mg_3_/Al-CO_3_ to *c* = 24.375 Å for the Mg_3_/Al-oxalate (in the case of short-long chains intercalation) and to *c* = 24.492 Å for the Mg_3_/Al_1_-4-biphenylacetonate (in the case of derivatives of aromatic hydrocarbons). These results led us to conclude that all anions studied have been successfully intercalated to the Mg_3_/Al LDHs structure.

The dimensions of anions (Table 2) show that the oxalate anion of intercalated LDH was the smallest by length (1.94 Å) and having the highest height (5.01 Å). Since the determined *c* parameter for the Mg_3_/Al-oxalate modified LDH is the largest between short-long chains intercalation, it can be deduced that the oxalate anion has specific vertical orientation in the LDHs (see Figure 3). In the case of aromatic hydrocarbons, the height of all anions is very similar. Therefore, the 4-biphenylacetonate which has a largest length (10.06 Å) has horizontal orientation in the LDH structure (Figure 3). The Mg_3_/Al-oxalate and the Mg_3_/Al-4-biphenylacetonate LDHs having similar basal spacings correspond to the intercalated LDHs with vertical and horizontal anion orientations in which they are grafting into the hydroxide layers. There are spherical energetic interferences between -CH_3_ groups of anions and M-OH hydroxide layers what cause difficult intercalation in the LDH structure. The formation of hydrogen bonds between water molecules in the layers, the hydroxide layers, the interlayer anions, and among the H_2_O molecules themselves is possible. The orientation of oxalate anion possibly is related to the formation of H_2_O molecules more compact structures with the two -COO^−^ groups than with the hydrophobic ends of the monocarboxylate. Four oxalate -COO- groups are distributed perpendicular to the layers, with two O-atoms coordinated to different hydroxide layers. In the case of 4-biphenylacetonate, the -COO- groups are orientated differently, and the O-atoms of its -COO^−^ groups that situated parallel to the layers can occupy M-OH sites along the H-H vectors, whereas those -COO- tend to occupy the centers of the M-OH triangles.

Europium substitution effects incorporating Eu^3+^ at the Al^3+^ positions in Mg_3_Al-organic anion LDHs have been investigated. According to [27], the Mg_3_/Al_0.99_Eu_0.01_ (with 1 mol% of Eu) have been prepared and intercalated with different organic anions. The XRD patterns (Figure 4 and Figure 5) for the hybrid inorganic-organic Mg_3_/Al_0.99_Eu_0.01_ LDHs showed, that the position of the (003) diffraction line is relevant to the interlayer distance and depends on the size of the intercalated organic anion. Surprisingly, the shift of the diffraction lines in the XRD patterns of intercalated with different organic anions of Eu^3+^-substituted LDHs is less pronounced in comparison with the samples without europium. This might be due to the reason, that the electrostatic attraction between mixed-metal cations and anions is weaker influencing on the distance of interlayer.

FT-IR spectra of Mg_3_/Al, Mg_3_/Al_0.99_Eu_0.01_ and hybrid inorganic-organic LDHs are shown in Figure 6 and Figure 7. The spectra of all samples are almost identical with very little differences. The broad absorptions visible at 3500–3000 cm^−1^ are characteristic vibrations of (-OH) groups [11]. The most intensive absorption bands detectible at 1360 cm^−1^ could be assigned to the asymmetric vibrations of CO_3_^2−^, which still exists in the interlayer of intercalated LDHs along with intercalated organic anions. The absorption bands in the range of 1570–1627 cm^−1^ are assigned to the of carbon-oxygen bonds of (-COO^−^) group. The first absorption band is related to the asymmetric vibration of the carboxylate group (υ_as_, COO^−^) and the second is attributable to the symmetric vibration of the carboxylate group (υ_s_, COO^−^), demonstrating the coordination of carboxylates to Mg_3_/Al-benzoate (h), Mg_3_/Al-1,3,5-benzentricarboxylate (i), Mg_3_/Al-4-methylbenzoate (Figure 6) and Mg_3_/Al_0.99_Eu_0.01_-benzoate (g), Mg_3_/Al_0.99_Eu_0.01_-1,3,5-benzentricarboxylate (h) (Figure 7). The absorption bands visible at 2850–2937 cm^−1^ are due to the C-H stretching vibrations of methylene (-CH_2_-) of the organic compounds. Thus, the FT-IR results prove the formation of the inorganic-organic hybrids and interactions of the introduced organic species with the LDH layers.

The emission spectra obtained at room temperature of Mg_3_/Al_0.99_Eu_0.01_ and Mg_3_/Al_0.99_Eu_0.01_ samples intercalated with benzoate, oxalate, laurate, malonate, succinate, tartrate, 1,3,5-benzentricarboxylate (BTC), 4-methylbenzoate (MB), 4-dimethylaminobenzoate (DMB) and 4-biphenylacetonate (BPhAc) anions (λ_ex_ = 394 nm) are presented in Figure 8. The emission spectra of Mg_3_/Al_0.99_Eu_0.01_—organic anion LDHs show four main emission lines between 550 nm and 740 nm. All observed emission bands are due to ^5^D_0_–^7^F_J_ (J = 1, 2, 3, 4) transitions of Eu^3+^ ions. According to the literature, the emissions are ^5^D_0_→^7^F_1_ (590 nm), ^5^D_0_→^7^F_2_ (613 nm), ^5^D_0_→^7^F_3_ (650 nm) and ^5^D_0_→^7^F_4_ (697 nm) transitions typical of Eu^3+^ ion [28]. The Eu^3+^ ions occupy a low-symmetry site, since the emission due to ^5^D_0_→^7^F_2_ transition is the strongest. Moreover, the results obtained indicate that the excitation energy to the Eu^3+^ ion in most of the cases is transferring from the organic anion ligands increasing the intensity of emission of the LDHs. Two mechanisms of intramolecular and intermolecular energy transfer between lanthanide ions and organic molecules have been suggested [29]. As was stated in [29], the intensity of emission of lanthanide distributed in host matrixes is affected by the energy matching degree between organic ligands and lanthanide ions. Evidently, when the energy matching degree is better, the energy transfer efficiency is higher and, consequently, the emission intensity of the compound is higher. The potency to absorb the UV radiation by interlayer organic anions and possible transfer this energy to the Eu^3+^ center by the interaction between the carboxyl oxygen of the intercalated anions with the hydrogen of the M(OH)_6_ octahedra via a hydrogen bond was suggested. The tartrate and benzoate having the strong basicity, showed higher ability to absorb the light [30]. Carbonate is a weaker base, thus transferring less energy to Eu^3+^ ions. The aromatic ring in the benzene can also influence the levels of resonant energy of lanthanide ions. As we can see from emission spectra the Mg_3_/Al_0.99_Eu_0.01_-tartrate and Mg_3_/Al_0.99_Eu_0.01_-benzoate LDHs show the highest emission intensity to compare with LDHs containing other organic ligands. Carboxylate and carbonyl groups connected with aromatic ring usually decrease the intensity of emission. The energy matching degree in the benzoate and Eu(III) complex obviously should be enhanced influencing the intensity of emission [31]. Moreover, the bridge of methylene groups -CH_2_- can break up the conjugated π-electron system [18].

The SEM micrographs depicted in Figure 9 represent the microstructure of Mg_3_/Al-CO_3_ and Mg_3_/Al_0.99_Eu_0.01_-CO_3_ layered double hydroxides. As seen, the solids are composed of particles having plate form and size about of 200–400 nm. The representative SEM micrographs of LDHs intercalated with different organic anions are shown in Figure 10. The surface microstructure still represents the characteristic features of LDHs [32], however, the particle sizes increased considerably (500–600 nm). Finally, the SEM micrographs of the samples which showed the most intensive emission Mg_3_/Al_0.99_Eu_0.01_-tartrate and Mg_3_/Al_0.99_Eu_0.01_-benzoate are presented in Figure 11. Evidently, the surface microstructure of these two samples is almost identical. The hexagonally shaped particles with the size of ~450–500 nm have formed.

## 4. Conclusions

Mg_3_/Al-CO_3_ and Mg_3_/Al_0.99_Eu_0.01_ LDHs intercalated with benzoate, oxalate, laurate, malonate, succinate, tartrate, 1,3,5-benzentricarboxylate (BTC), 4-methylbenzoate (MB), 4-dimethylaminobenzoate (DMB) and 4-biphenylacetonate (BPhAc) were prepared by sol-gel processing. The XRD analysis results clearly showed that the positions of diffraction peaks (003) of LDHs intercalated with anions were shifted to smaller 2θ angle values. However, the shift of the diffraction lines in the XRD patterns of intercalated with different organic anions of Eu^3+^-substituted LDHs was less pronounced in comparison with the samples without europium. The FT-IR results demonstrated once again the formation of the inorganic-organic hybrids and interaction of the organic ions with the LDH layers. The obtained results let us to conclude that depending on the size of anions these species could have specific vertical or horizontal orientations in the LDH structure. The microstructure of Mg_3_/Al-CO_3_, Mg_3_/Al_0.99_Eu_0.01_-CO_3_ and Mg_3_/Al_0.99_Eu_0.01_-organic anion was typical for LDH samples. The SEM images showed the formation of hexagonally shaped plate-like particles of LDHs of 200–600 nm in size with high degree of agglomeration. The room temperature luminescence of Mg_3_/Al_0.99_Eu_0.01_ and Mg_3_/Al_0.99_Eu_0.01_ samples intercalated with benzoate, oxalate, laurate, malonate, succinate, tartrate, 1,3,5-benzentricarboxylate (BTC), 4-methylbenzoate (MB), 4-dimethylaminobenzoate (DMB) and 4-biphenylacetonate (BPhAc) anions under excitation at 394 nm was investigated. In all spectra, the typical four emission bands due to transitions of ^5^D_0_→^7^F_1_ (590 nm), ^5^D_0_→^7^F_2_ (613 nm), ^5^D_0_→^7^F_3_ (650 nm) and ^5^D_0_→^7^F_4_ (697 nm) of Eu^3+^ ion were determined. The Mg_3_/Al_0.99_Eu_0.01_-tartrate and Mg_3_/Al_0.99_Eu_0.01_-benzoate LDHs showed the highest emission intensity to compare with LDHs containing other organic ligands.

## Figures and Tables

**Figure 1 materials-12-00736-f001:**
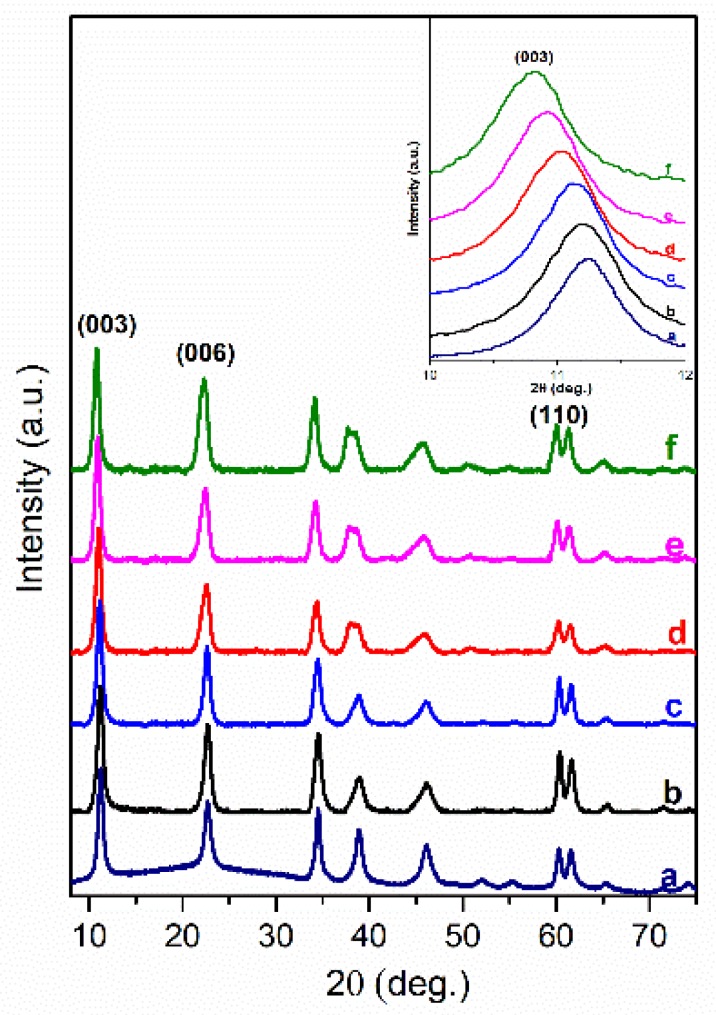
XRD patterns of Mg_3_/Al-CO_3_ layered double hydroxides (LDH) (**a**) and Mg_3_/Al-CO_3_ intercalated with organic anions: Mg_3_/Al-succinate (**b**), Mg_3_/Al-malonate (**c**), Mg_3_/Al-tartrate (**d**), Mg_3_/Al-laurate (**e**), and Mg_3_/Al-oxalate (**f**).

**Figure 2 materials-12-00736-f002:**
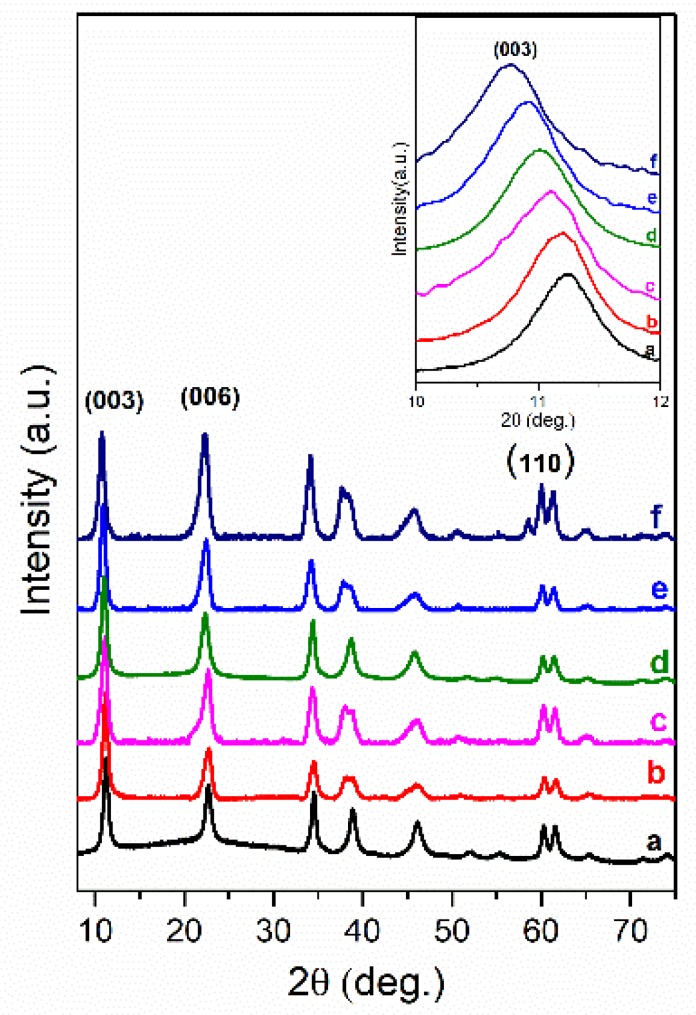
XRD patterns of Mg_3_/Al-CO_3_ LDH (**a**) and Mg_3_/Al-CO_3_ intercalated with organic anions Mg_3_/Al-4-dimethylaminobenzoate (**b**), Mg_3_/Al-4-methylbenzoate (**c**), Mg_3_/Al-1,3,5-benzentricarboxylate (**d**), Mg_3_/Al-benzoate (**e**) and Mg_3_/Al-4-biphenylacetonate (**f**).

**Figure 3 materials-12-00736-f003:**
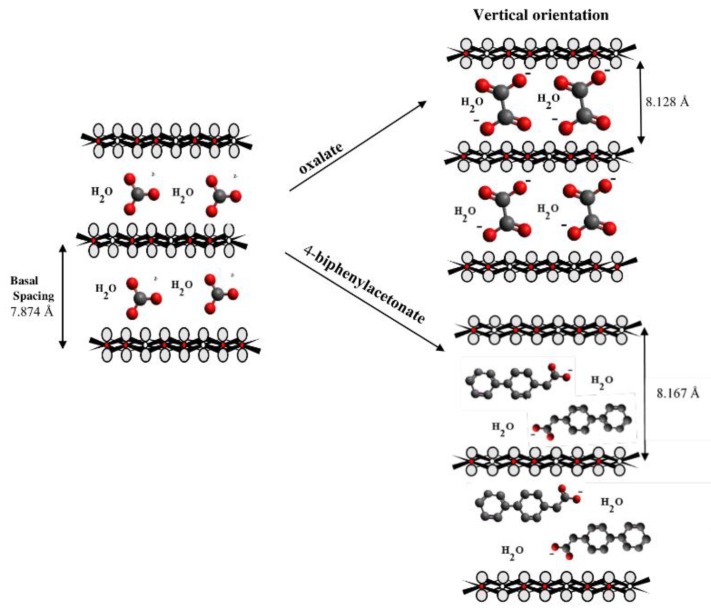
A schematic structure of LDHs with interlayer carbonate anion and the specific orientation of oxalate and 4-biphenylacetonate anions between the layers.

**Figure 4 materials-12-00736-f004:**
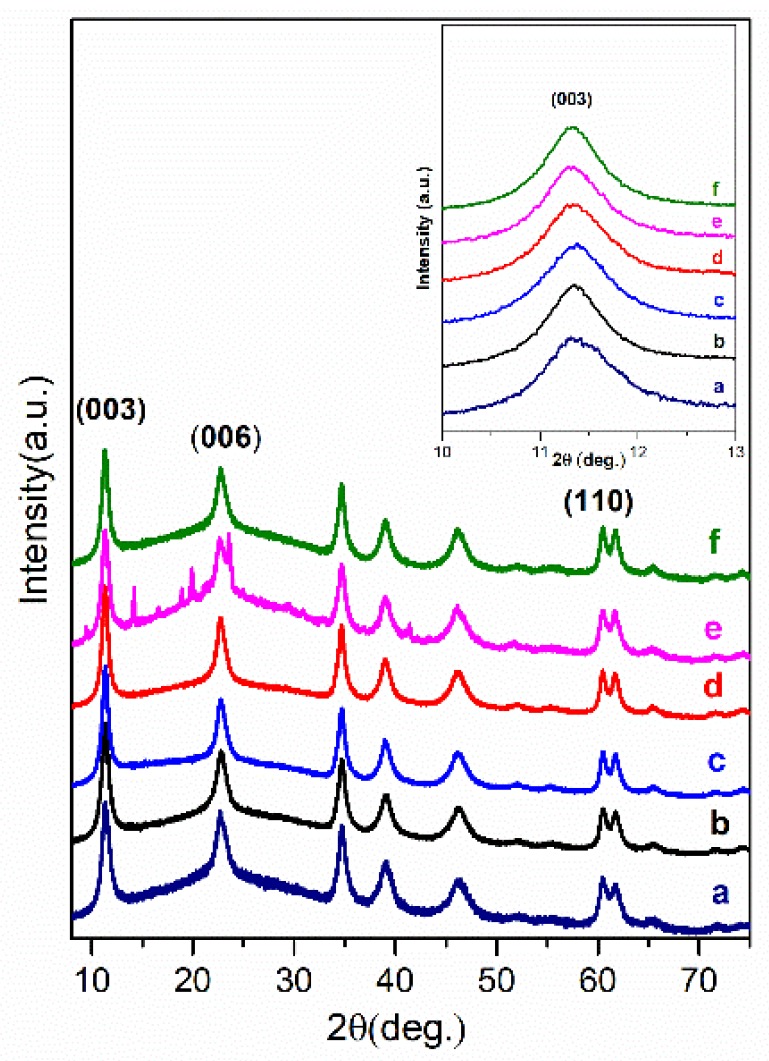
XRD patterns of Mg_3_/Al_0.99_Eu_0.01_-CO_3_ (**a**) and hybrid inorganic-organic LDHs: Mg_3_/Al_0.99_Eu_0.01_-succinate (**b**), Mg_3_/Al_0.99_Eu_0.01_-malonate (**c**), Mg_3_/Al_0.99_Eu_0.01_-tartrate (**d**), Mg_3_/Al_0.99_Eu_0.01_-laurate (**e**) and Mg_3_/Al_0.99_Eu_0.01_-oxalate (**f**).

**Figure 5 materials-12-00736-f005:**
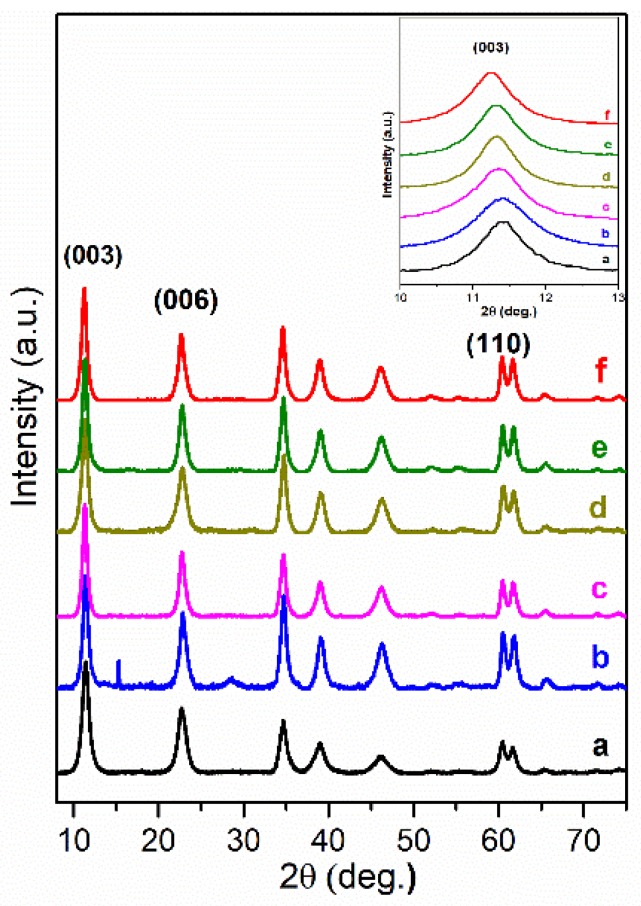
XRD patterns of Mg_3_/Al_0.99_Eu_0.01_-CO_3_ (**a**) and hybrid inorganic-organic LDHs: Mg_3_/Al_0.99_Eu_0.01_-4-dimethylaminobenzoate (**b**), Mg_3_/Al_0.99_Eu_0.01_-4-methylbenzoate (**c**), Mg_3_/Al_0.99_Eu_0.01_-1,3,5-benzentricarboxylate (**d**), Mg_3_/Al_0.99_Eu_0.01_-benzoate (**e**) and Mg_3_/Al_0.99_Eu_0.01_-4-biphenylacetonate (**f**).

**Figure 6 materials-12-00736-f006:**
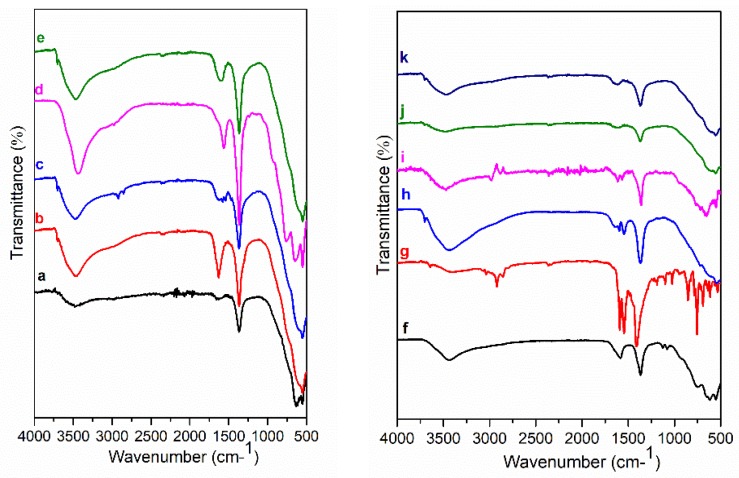
FT-IR spectra of Mg_3_/Al-CO_3_ (**a**) and hybrid inorganic-organic LDHs: Mg_3_/Al-oxalate (**b**), Mg_3_/Al-laurate (**c**), Mg_3_/Al-succinate (**d**), Mg_3_/Al-malonate (**e**). FT-IR spectra of Mg_3_/Al-tartrate (**f**), Mg_3_/Al-4-biphenylacetonate (**g**), Mg_3_/Al-benzoate (**h**), Mg_3_/Al-1,3,5-benzentricarboxylate (**i**), Mg_3_/Al-4-methylbenzoate (**j**) and Mg_3_/Al-4-dimethylaminobenzoate (**k**).

**Figure 7 materials-12-00736-f007:**
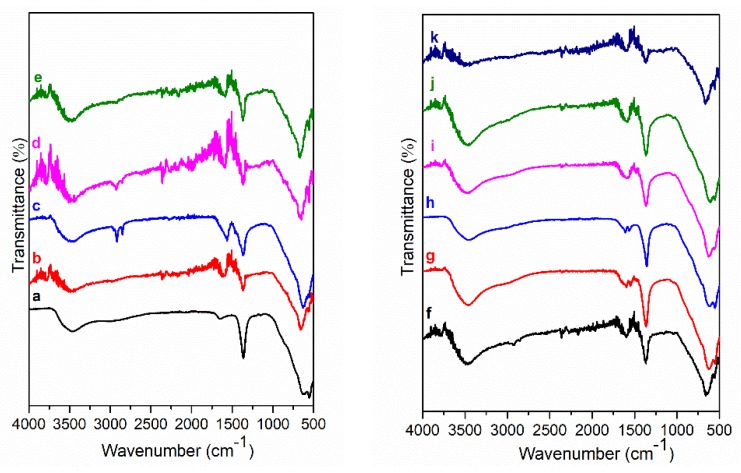
FT-IR spectra of Mg_3_/Al_0.99_Eu_0.01_-CO_3_ (**a**) and hybrid inorganic-organic LDHs: Mg_3_/Al_0.99_Eu_0.01_-oxalate (**b**), Mg_3_/Al_0.99_Eu_0.01_-laurate (**c**), Mg_3_/Al_0.99_Eu_0.01_-succinate (**d**), Mg_3_/Al_0.99_Eu_0.01_-malonate (**e**). FT-IR spectra of Mg_3_/Al_0.99_Eu_0.01_-tartrate (**f**), Mg_3_/Al_0.99_Eu_0.01_-benzoate (**g**), Mg_3_/Al_0.99_Eu_0.01_-1,3,5-benzentricarboxylate (**h**), Mg_3_/Al_0.99_Eu_0.01_-1-4-biphenylacetonate (**i**), Mg_3_/Al_0.99_Eu_0.01_-4-methylbenzoate (**j**) and Mg_3_/Al_0.99_Eu_0.01_-4-dimethylaminobenzoate (**k**).

**Figure 8 materials-12-00736-f008:**
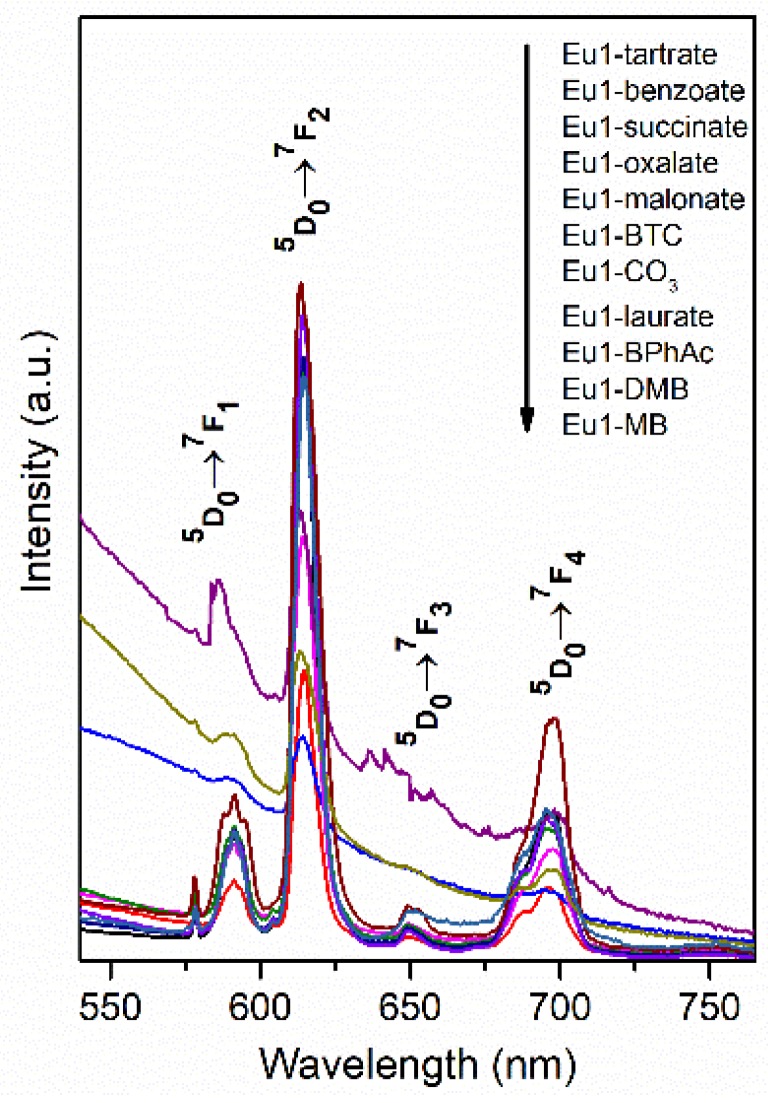
Emission spectra of Mg_3_/Al_0.99_Eu_0.01_ and LDHs intercalated with tartrate, benzoate, succinate, oxalate, malonate, 1,3,5-benzentricarboxylate, laurate, 4-biphenylacetonate, 4-dimethylaminobenzoate and 4-methylbenzoate. (Ex = 394 nm).

**Figure 9 materials-12-00736-f009:**
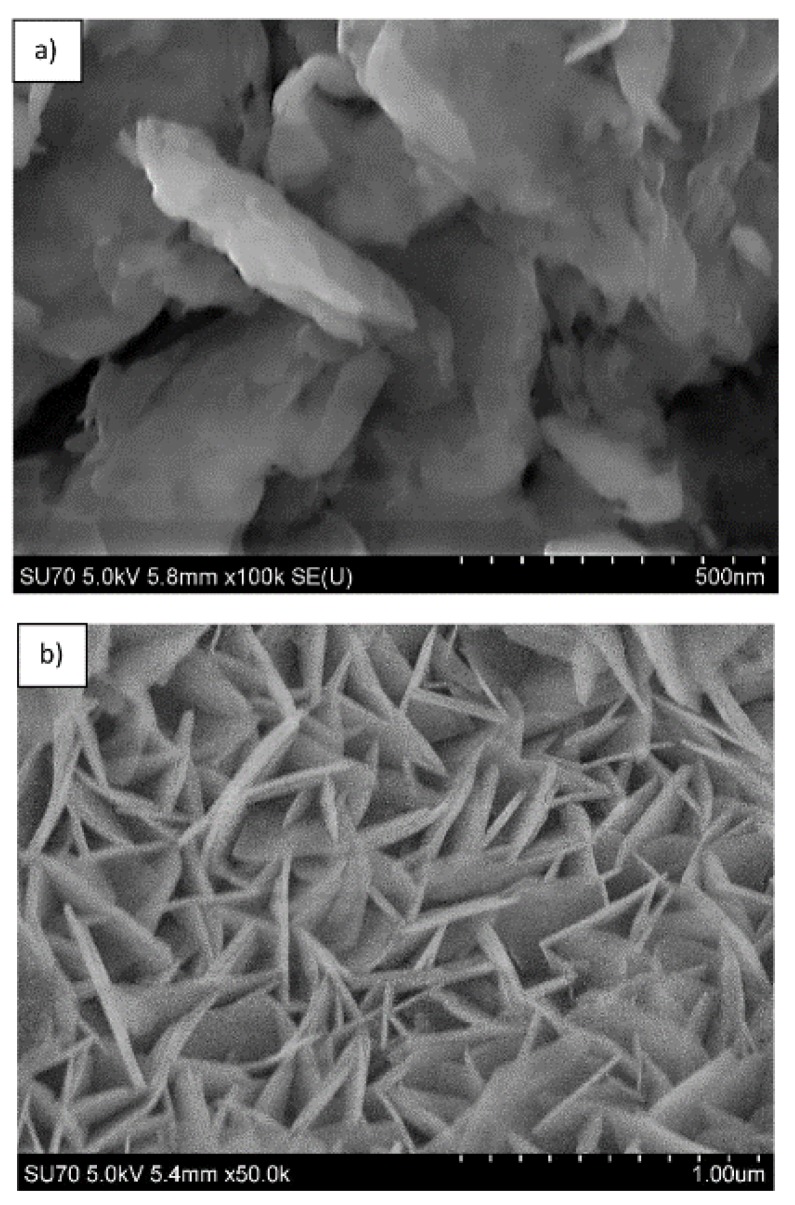
SEM micrographs of Mg_3_/Al-CO_3_ (**a**) and Mg_3_/Al_0.99_Eu_0.01_-CO_3_ (**b**) LDHs.

**Figure 10 materials-12-00736-f010:**
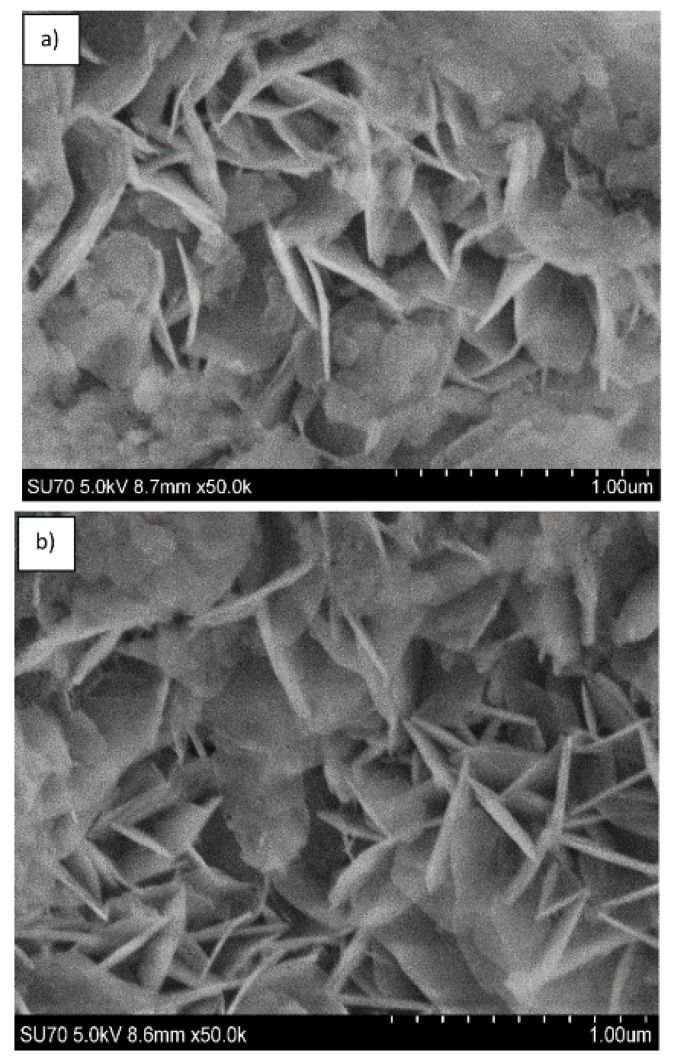
SEM micrographs of Mg_3_/Al-oxalate (**a**) and Mg_3_/Al-4-biphenylacetonate (**b**) LDHs.

**Figure 11 materials-12-00736-f011:**
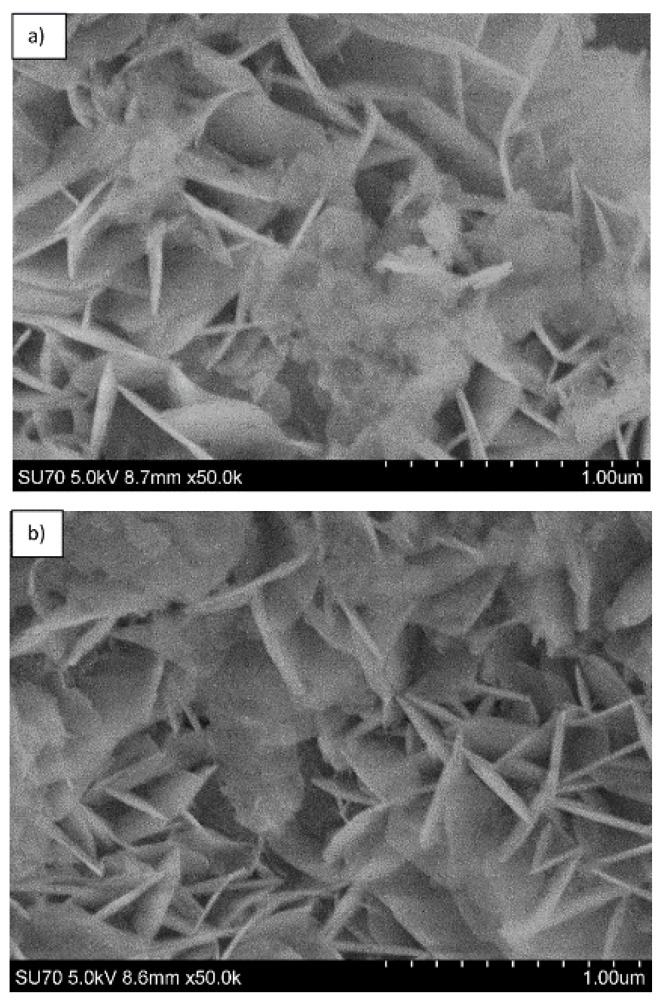
SEM micrographs of Mg_3_/Al_0.99_Eu_0.01_-tartrate (**a**) and Mg_3_/Al_0.99_Eu_0.01_-benzoate (**b**) LDHs.

**Table 1 materials-12-00736-t001:** The determined values of *d* spacing and lattice parameters of anion-intercalated Mg_3_/Al LDHs. The standard deviations for all measurements do not exceed ±0.0005.

Sample	Basal Spacing/Å		Cell Parameter/Å	
	d(003)	d(110)	a	c
Mg_3_/Al-CO_3_	7.8744	1.5350	3.068	23.613
Mg_3_/Al-oxalate	8.1286	1.5385	3.076	24.375
Mg_3_/Al-laurate	8.0905	1.5380	3.075	24.261
Mg_3_/Al-tartarate	7.9970	1.5359	3.070	23.981
Mg_3_/Al-malonate	7.9568	1.5343	3.067	23.860
Mg_3_/Al-succinate	7.9454	1.5333	3.065	23.826
Mg_3_/Al-4-biphenylacetonate	8.1675	1.5396	3.078	24.492
Mg_3_/Al-benzoate	8.0875	1.5384	3.075	24.252
Mg_3_/Al-4-methylbenzoate	8.0564	1.5383	3.075	24.159
Mg_3_/Al-1,3,5-benzentricarboxylate	8.0328	1.5373	3.073	24.088
Mg_3_/Al-4-dimethylaminobenzoate	7.8907	1.5324	3.063	23.662

**Table 2 materials-12-00736-t002:** Formula and dimensions of anions.

Anion	Chemical Formula	Structural Formula and Dimensions
Oxalate	(C_2_H_4_)^2−^	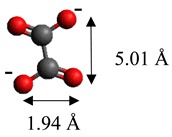
Laurate	(C_12_H_23_O_2_)^2−^	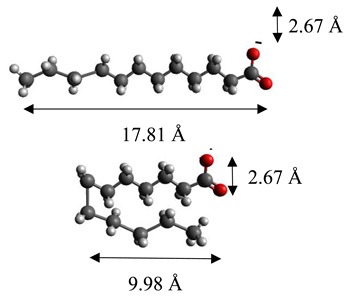
Malonate	(C_3_H_2_O_4_)^2−^	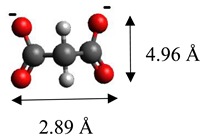
Succinate	(C_3_H_4_O_4_)^2−^	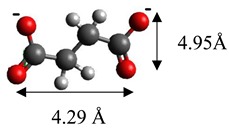
Tartrate	(C_4_H_4_O_6_)^2−^	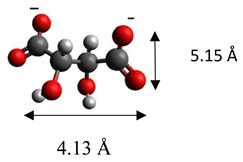
Benzoate	(C_7_H_5_O_2_)^−^	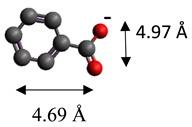
1,3,5-benzentricarboxylate	(C_9_H_5_O_6_)^3−^	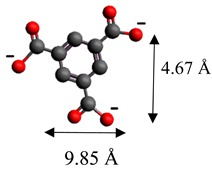
4-methylbenzoate	(C_8_H_7_O_2_)^−^	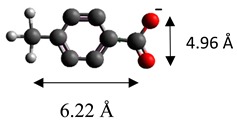
4-dimethylaminobenzoate	(C_9_H_10_O_2_)^−^	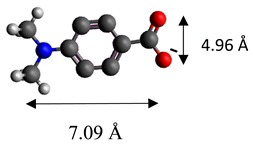
4-biphenylacetonate	(C_14_H_11_O_2_)^−^	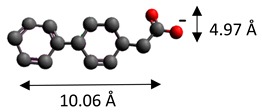

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
