# Peer review of "Undoped and Eu3+ Doped Magnesium-Aluminium Layered Double Hydroxides: Peculiarities of Intercalation of Organic Anions and Investigation of Luminescence Properties"

_materials, 2019, doi:10.3390/ma12050736_

Reviewer 1 Report

See attached file.

Author Response

Reviewer #1:

1.     Use “Mg3/Al” instead of “Mg3/Al1” in the manuscript.

      Answer:

      In whole text the abbreviation “Mg3/Al1” has been corrected to “Mg3/Al”.

2.     Change “Mg3/l0.99Eu0.01” to “Mg3/Al0.99Eu0.01” in line 13.

      Answer:

       It has been changed to “Mg3/Al0.99Eu0.01” in line 13.

3. Interaction between LDHs and carbonate or carboxylate-containing substances is an important topic due to the high affinity of these types of anions to the LDH surface. I agree that intercalation may occur especially with anions of smaller size. However, the adsorption of these anions on the outer surface of LDHs cannot be neglected. In addition, key publications (e.g., Langmuir, 2015, 31, 12609-12617, J. Colloid Interface Sci., 2008, 326, 522-529 and Soft Matter, 2016, 12, 4024-4033), which discusses the significant adsorption of carbonate anions or carboxylate-containing oligomers and polymers on the outer surface of LDHs are neither cited nor discussed in the manuscript. I strongly recommend extending the introduction as well as the results part with the possible adsorption on the outer surface. The above-mentioned papers should be cited. This topic was also discussed in a recent review: ChemPlusChem, 2017, 82, 121-131. I think that this is the weakest point of the manuscript.

        Answer:

         All suggested articles have been added and citied in the introduction part; lines 44 and 56.

4.     The SO42- > CO32- order mentioned in line 36 is wrong and should be the opposite (see reference 3).

Answer:

The order has been corrected.

5.     The resolution quality of the tables and figures are very poor.

Answer:

The resolution quality of the tables and figures has been improved.

6.     The authors assume the orientation of the intercalated anions on the basis of basal spacing determined from the (003) peak. However, this spacing is equal to the interlayer distance plus the thickness of one layer. If one subtracts the thickness of one layer from the basal spacing, the orientation of oxalate shown in Figure 3 is not possible (see Clay Clay Miner. 23 (1975) 369).

Answer:

It has been corrected to two layers in Figure 3.

 Reviewer 2 Report

The manuscript reports on the synthesis of a series of Mg/Al and Mg/Al-Eu Layered double hydroxide=des (LDH) intercalated with various alkyl or aryl carboxylates. The purpose is to investigate the effect of the organic host on the luminescence of Eu3+ ions. This work is a complement to previous works on Tb3+ or Eu3+ doped LDH. The basic idea is interesting, yet the present work does not provide real progress in the understanding of properties of these hybrid systems. Several points have to addressed before any publication.

1)      To be precise, LDH precursors are synthesized by the sol-gel method, but LDH are synthesized by reformation from oxides in water.

 2)      Apart general comments on the structural features of carbonated or nitrated LDH, the discussion of the structures of the title compounds is not convincing.

 Actually, the interlayer distances (d003) in the un-doped series varies very weakly and standard deviations on distances and parameters in table 1 should be given.

The tentative discussion on the orientation of the guest molecules based only on the interlayer distance is a bit speculative. Indeed, some molecules are not that anisotropic in size, some are mono-anions other bis-anions with peculiar interaction with the cationic layers. Flat arrangement or tilt angle can also occur, depending on the charge density within the host layers, as well as single of double layer, possibly interpenetrated, arrangement.

Detailed discussion of possible structural model would need additional data, as 1D Fourier analysis, PDF analysis, or coupled FTIR/Raman spectroscopy.

The chemical composition including water and CO3 ions should be given to evaluate charge density and anion packing possibilities.

In addition, the measurements in table 2 are not precisely drawn. Several double arrows do not fit molecular moieties and the thickness of flat molecules is not considered.

 3)      Regarding FTIR, the scale in transmittance is too small to observe small features. The organic guests are expected to exhibit characteristic bands.

The FTIR spectra of anions alone should be given to compare with that of the LDH compounds. A special focus on the carboxylate asym and sym bands would give valuable information.

It is noticed in Fig 6 g that a set of bands merely due to benzoate are not observed in the Eu doped analogue in fig 7 g. Again, chemical analysis is important here.

 4)      The discussion of the luminescence is short. The data are limited to emission. Absorption and excitation spectra should be analysed to investigated more deeply the photoluminescence and possible ligand to Eu transfer.

Hence, as written in the conclusion, this work is essentially in keeping with what is expected from previous works with no real novelty.

For all reasons above, I suggest rejection of this paper.

Author Response

Reviewer #2:

 1.     To be precise, LDH precursors are synthesized by the sol-gel method, but LDH are synthesized by reformation from oxides in water.

Answer:

Yes, we can agree with this.

 2.     Apart general comments on the structural features of carbonated or nitrated LDH, the discussion of the structures of the title compounds is not convincing.

We cannot agree with this.

 3.     Actually, the interlayer distances (d003) in the un-doped series varies very weakly and standard deviations on distances and parameters in table 1 should be given.

Answer:

The standard deviations for all measurements do not exceed                                               0.0005. It has been added to Table 1.

 4.     The tentative discussion on the orientation of the guest molecules based only on the interlayer distance is a bit speculative. Indeed, some molecules are not that anisotropic in size, some are mono-anions other bis-anions with peculiar interaction with the cationic layers. Flat arrangement or tilt angle can also occur, depending on the charge density within the host layers, as well as single of double layer, possibly interpenetrated, arrangement. Detailed discussion of possible structural model would need additional data, as 1D Fourier analysis, PDF analysis, or coupled FTIR/Raman spectroscopy.

           Answer:

           We agree with Reviewer that discussion about the orientation of the guest molecules is slightly speculative. However, the used this approximate                      assumption model in our study.

 5.     The chemical composition including water and CO3 ions should be given to evaluate charge density and anion packing possibilities.

Answer:

The general formula for LDH is [MII 1-x MIII x (OH)2 ] x+ (Am-)x/m]•nH2O and it indicates that it is possible to synthesize a number of compounds with different stoichiometries. In the natural hydrotalcites the value of x is generally equal to 0.25. Thus, the XRF results clearly confirmed that it is possible to synthesize LDH with the above formula having more than two metals [F. Cavani, F. Trifiro, A. Vaccari, Catal. Today 11, 173 (1991)]. There are, however, many difficulties in determining the exact value of x in LDH. An elemental analysis of the metal content of a solid phase will not give correct values if the LDH is not monophasic, i.e., mixed with MII(OH)2, MIII(OH)3 /MIIIOOH or other phases which could segregate when the synthesis mixture contains either very high or very low MII/MIII ratios. However, more often these phases are amorphous and cannot be detected by XRD [D.G. Evans, R.C.T. Slade, In: D.M.P. Mingos (Ed.), (Structure and Bonding, Springer-Verlag, Berlin, Heidelberg, 2006)]. The obtained x values of synthesized LDH by XRF technique are in the expected range. They are not lower than 0.2 and not higher than 0.3. So, the results of XRF analysis let us conclude that single-phase layered double hydroxides have formed during sol-gel synthesis. The carbonate content in synthesized samples was calculated from the MII/MIII atomic ratios, assuming that carbonate is the only charge balancing interlayer anion. The water content in the formula was determined from the results of TG analyses. The chemical composition was defined to be [Mg0.75Al0.25(OH)2] (CO3)0.125 ·4H2O and it has been represented in to the characterization part line 112.

6.     In addition, the measurements in table 2 are not precisely drawn. Several double arrows do not fit molecular moieties and the thickness of flat molecules is not considered.

Answer:

It has been corrected.

 7.     Regarding FTIR, the scale in transmittance is too small to observe small features. The organic guests are expected to exhibit characteristic bands.

Answer:

The FTIR results has been separated and improved to Figure 6a and Figure 6b, Figure 7a and Figure 7b.

  8.     The FTIR spectra of anions alone should be given to compare with that of the LDH compounds. A special focus on the carboxylate asym and sym bands would give valuable information.

Answer:

In our opinion this information would be redundant.

 9.     It is noticed in Fig 6 g that a set of bands merely due to benzoate are not observed in the Eu doped analogue in fig 7 g. Again, chemical analysis is important here.

Answer:

The FTIR spectra analysis has been improved line 202.

 10.   The discussion of the luminescence is short. The data are limited to emission. Absorption and excitation spectra should be analysed to investigated more deeply the photoluminescence and possible ligand to Eu transfer.

Answer:

There has been made some corrections, see line 332 and 348.

 Round  2

Reviewer 1 Report

The authors answered my comments, I recommend publishing the manuscript as is.